# Hepatic Failure in COVID-19: Is Iron Overload the Dangerous Trigger?

**DOI:** 10.3390/cells10051103

**Published:** 2021-05-04

**Authors:** Franca Del Nonno, Roberta Nardacci, Daniele Colombo, Ubaldo Visco-Comandini, Stefania Cicalini, Andrea Antinori, Luisa Marchioni, Gianpiero D’Offizi, Mauro Piacentini, Laura Falasca

**Affiliations:** 1Pathology Unit, National Institute for Infectious Diseases “Lazzaro Spallanzani”-IRCCS, 00149 Rome, Italy; franca.delnonno@inmi.it (F.D.N.); daniele.colombo@inmi.it (D.C.); 2Laboratory of Electron Microscopy, National Institute for Infectious Diseases “Lazzaro Spallanzani”-IRCCS, 00149 Rome, Italy; roberta.nardacci@inmi.it (R.N.); mauro.piacentini@uniroma2.it (M.P.); 3Hepatology Unit “Lazzaoro Spallanzani”-IRCCS, 00149 Rome, Italy; Ubaldo.viscocomandini@inmi.it (U.V.-C.); gianpiero.doffizi@inmi.it (G.D.); 4HIV/AIDS Department, National Institute for Infectious Diseases “Lazzaro Spallanzani”-IRCCS, 00149 Rome, Italy; stefania.cicalini@inmi.it (S.C.); andrea.antinori@inmi.it (A.A.); 5Clinical Department, National Institute for Infectious Diseases “Lazzaro Spallanzani”-IRCCS, 00149 Rome, Italy; luisa.marchioni@inmi.it; 6Department of Biology, University of Rome “Tor Vergata”, 00133 Rome, Italy

**Keywords:** SARS-CoV-2, COVID-19, liver, iron overload, ferritin, electron microscopy, mitochondria, lipid droplets

## Abstract

Liver injury in COVID-19 patients has progressively emerged, even in those without a history of liver disease, yet the mechanism of liver pathogenicity is still controversial. COVID-19 is frequently associated with increased serum ferritin levels, and hyperferritinemia was shown to correlate with illness severity. The liver is the major site for iron storage, and conditions of iron overload have been established to have a pathogenic role in development of liver diseases. We presented here six patients who developed severe COVID-19, with biochemical evidence of liver failure. Three cases were survived patients, who underwent liver biopsy; the other three were deceased patients, who were autopsied. None of the patients suffered underlying liver pathologies. Histopathological and ultrastructural analyses were performed. The most striking finding we demonstrated in all patients was iron accumulation into hepatocytes, associated with degenerative changes. Abundant ferritin particles were found enclosed in siderosomes, and large aggregates of hemosiderin were found, often in close contact with damaged mitochondria. Iron-caused oxidative stress may be responsible for mitochondria metabolic dysfunction. In agreement with this, association between mitochondria and lipid droplets was also found. Overall, our data suggest that hepatic iron overload could be the pathogenic trigger of liver injury associated to COVID-19.

## 1. Introduction

SARS-CoV-2 causes a wide spectrum of illness ranging from mild symptoms to severe presentation, and death. Usually COVID-19 causes pneumonia; however, clinical and epidemiological studies have established that COVID-19 presents as a spectrum of clinical manifestations [1,2], and studies performed on whole-body autopsies in patients who died of COVID-19 revealed that SARS-CoV-2 infection causes significant alterations in most body organs [3]. In patients with COVID-19, symptoms of gastrointestinal and hepatic involvement have been progressively recognized [4]. Derangement of ALT/AST levels represents the main indication of liver damage in COVID-19, accompanied by slightly elevated bilirubin levels. The incidence of hepatic injury ranges from 14.8% to 53%, but in severe COVID-19 patients the incidence of liver injury may increase to 58–78% [5,6]. Different mechanisms have been proposed that may be responsible for hepatic injury: SARS-CoV-2 direct damage on hepatocytes and biliary epithelium, or indirect damage induced by exaggerated cytokines storm, and/or drug-induced hepatoxicity. Nevertheless, the exact cause of liver injury has not yet been clarified [7].

Hyperferritinemia has been recognized as a hallmark of severe COVID-19 [8,9]. Excess iron is toxic and can damage tissues by free radical formation and lipid peroxidation. For this reason, the systemic iron levels should be strictly maintained. Ferritin is the most relevant cellular iron storage protein. The liver exerts a pivotal role in iron homeostasis and is the main organ for the iron storage as ferritin complex. Conditions of iron overload are established to be implicated in liver injury. Hereditary hemochromatosis is the classical example of a liver disease caused by iron [10], but iron-catalyzed oxidative damage plays a key role in the development of various forms of liver diseases [11,12].

This study aimed to explore a possible link between hepatic failure and iron toxicity in the course of COVID-19. Histopathological and ultrastructural changes were deeply investigated in the liver of COVID-19 survived and deceased patients who presented biochemical evidence of liver injury. 

## 2. Materials and Methods

### 2.1. Study Cohort

Cases considered for the study included three COVID-19 patients, who were treated and deceased at the National Institute for Infectious Diseases (INMI) “Lazzaro Spallanzani” IRCCS Hospital (Rome, Italy), and three COVID-19 patients extracted from case records of patients who died and underwent complete post-mortem examination at INMI Lazzaro Spallanzani-IRCCS Hospital. The study was approved by the Institutional Ethic Board (IEB) of the National Institute for Infectious Diseases “L. Spallanzani” (approval number: 9/2020). All data were anonymized to protect the confidentiality of individual participants.

Exclusion criteria for cases were HCV chronicity, positive serology for hepatitis B and/or human immunodeficiency virus, history of liver transplant, alcoholic liver disease and non-alcoholic fatty liver disease, or patients with other liver diseases. Alcohol or recreational drugs were also considered as exclusion criteria.

### 2.2. Liver Biopsies

Three confirmed cases of COVID-19 patients, admitted and treated at the National Institute for Infectious Diseases (INMI) “Lazzaro Spallanzani” in Rome, who developed hepatic failure during hospitalization, were enrolled for this study. Liver samples were obtained by liver biopsy performed on clinical indications in order to stage the liver injury.

### 2.3. Autoptic Liver

Liver tissue samples were obtained from complete post-mortem examination of three SARS-CoV-2 infected patients. All patients were diagnosed with COVID-19 by PCR-test for SARS-CoV-2 (using RealStar® SARS-CoV-2 RT-PCR Kit 1.0 (Altona Diagnostic GmbH)) performed on nasopharyngeal swab and/or on autoptic samples. 

Autopsies were performed according to guidance for post-mortem collection and submission of specimens and biosafety practices [3] to reduce the risk of transmission of infectious pathogens during and after the post-mortem examination. 

Cases presenting severe hepatic autolysis and/or patients who died on arrival, which had either no or limited clinical data, were excluded from this study.

### 2.4. Clinical Evaluation

Patients were analyzed considering clinical features, laboratory parameters, and length of hospital stay. Abnormal liver function was defined as increased levels of alanine aminotransferase (ALT), alanine aspartate aminotransferase (AST), gamma-glutamyl transferase (GGT), and total bilirubin.

### 2.5. Histological Analyses

Liver specimens from liver tissues were fixed in 10% neutral-buffered formalin and routinely processed to paraffin blocks. Sections of tissues (4 μm) were stained with hematoxylin and eosin (H&E), with periodic acid–Schiff–diastase (PAS-D), with Masson’s trichrome for collagen fibers to assess of liver fibrosis, and Perl's Prussian blue to highlight iron deposition. Each biopsy sample was evaluated according to the grading system for inflammation and necrosis, and according the staging system for fibrosis [13]. The grading of liver steatosis was based on the percentage of fat within the hepatocytes: grade 0 (healthy, <5%), grade 1 (mild, 5–33%), grade 2 (moderate, 34–66%), and grade 3 (severe, >66%). Hepatic iron was graded on a semi-quantitative scale in increasing order of severity, with 0 being absent, and 4 representing intensely clumped iron within the cells, according to Scheuer’s classification [14]. Quantification was performed by counting positive cells in 7–8 different fields at 200× magnification. Pathological assessment was performed by two pathologists who were blinded to the patient data. 

### 2.6. Immunohistochemical Anlysis

Deparaffinized and rehydrated sections were used for immunohistochemistry. Samples were immersed in 10 mM sodium citrate pH 6.0, microwaved for antigen retrieval, and stained on the BenchMark ULTRA system fully automated instrument (Roche, Tucson, AZ 85755, USA) with antibodies directed against CD4 (Ventana SP35), CD8 (Ventana SP57), CD68 (Ventana KP-1), CD34 (Ventana QBEnd/10), vWF (Leica 36B11), aSMA (Ventana 760-2833), Coronavirus SARS-CoV Nucleoprotein (Sino Biological 40143-T62), and SARS Nucleocapsid protein (Novusbio NB100-56576). Immune-histochemical quantification was performed by light microscopy, counting the number of positive cells per at least seven different high-power fields (HPF, 40× magnification). Results were expressed as the mean of the values obtained from all patients and presented as mean + SD. Negative control stainings were performed by omitting the primary antibodies. In addition, sections of control liver, obtained from formalin-fixed and paraffin-embedded blocks of retrospective/archival samples (i.e., patient selection and liver biopsy were not for the purpose of the study) were used. Cases considered included one of primary sclerosing cholangitis, and one minimal hepatitis after HBV infection.

## 3. Results

### 3.1. Patient Demographics and Clinical Characteristics

Demographic, medical history, and clinical features of our cohort of patients are summarized in Table 1 and Table 2. The three biopsied patients were male, with a median age of 42 years (range 40–46 years) (Table 1).

One of these patients was considered obese (BMI 33 kg/m^2^); the others had no co-morbidities, or any cause of immunosuppression, and were healthy prior to hospital admission. One of these patients had a long stay in the ICU and developed critical illness. Ages of the deceased patients ranged between 63 and 75 years; the median age was 69 years. Two of the patients were male and one was female (Table 2).

Laboratory data obtained during hospital stay are shown in Table 3. Fibrinogen and D-dimer levels were elevated already at admission. D-dimer remained elevated in survived patients even at discharge.

Liver enzymes, and AST and ALT levels, initially abnormal in three cases, increased in all patients during hospitalization. Ferritin levels were elevated in all patients (peak range 670–3292 μg/L). Some patients showed a cholestatic pattern (mild to moderate elevation of gamma glutamyl transpeptidase and direct bilirubin levels) (Table 3).

### 3.2. Liver Pathological Features in Survived Patients

Main pathologic findings are summarized in Table 4.

Histological analysis of liver biopsies showed mild lobular and portal inflammatory infiltrates (Figure 1A,B). Macrovesicular steatosis (Figure 1A) was found in all three patients (>33% in 2 patients; >5% in 1 patient), even in the absence of previous underlying conditions. The presence of ballooning degeneration, associated with lobular spotty necrosis and rare acidophil bodies, was observed (Figure 1B); no Mallory–Denk bodies were found. Kupffer cell hyperplasia was always observed, as demonstrated by PAS-D+ staining (data not shown). Sinusoidal dilatation mainly in centrilobular areas was present in two patients (Figure 1C). Fibrous expansion of portal areas with short fibrous septa was found only in the obese patient (Figure 1D). Infiltrating inflammatory cells (Figure 2) mainly consisted in T-lymphocytes, CD8+ (10.99 ± 2.18 CD8+ cells/field vs 3.39 ± 0.92 CD4+ cells/field), mixed with CD68+ macrophages (4.25 ± 0.36 cells/field) (Figure 2A,B). Immunohistochemical staining for CD68 also labelled Kupffer cells as sinusoids, recognizable for the star-shaped morphology (Figure 2B). Immunohistochemical staining for the vascular marker CD34 showed a positive reaction not only in the inflamed portal tract, but also extending beyond the periportal areas, suggesting the occurrence of an angiogenic process (Figure 2C,D). Alteration of vascular structures was further demonstrated by immune-staining of alpha-smooth muscle actin (αSMA) and von Willebrand factor (vWF) (Appendix A) as compared to control livers from non-COVID-19 patients (Appendix A).

### 3.3. Liver Pathological Features in Deceased Patients

Histopathologic abnormalities observed in livers from deceased patients are summarized in Table 4. Severe macrovacuolar steatosis was the most common finding in two patients (Figure 1E,F). Hepatic sinusoidal dilatation was detected in all patients (Table 4), and extravasation of red blood cells into the Disse’s space was sometimes observed (Figure 1F). In two cases, this condition was associated with small vein congestion and zonal necrosis of the hepatocytes in the periportal area. Portal inflammation was mild in all cases, and lobular inflammation was focal in two patients (Table 4). Inflammatory cells, mainly found in the portal area, mostly consisted of CD8+ T lymphocytes (Figure 2E). Immunohistochemical labelling for CD34 showed positive staining of the portal tract vasculature and extending beyond the periportal regions, with centrizonal microvessels and sinusoids CD34+ positivity, indicative of vascular remodeling (Figure 2F).

### 3.4. Iron Deposition in the Livers of COVID-19 Patients

The presence of iron accumulation was histologically evaluated by Perl’s staining. The stain highlights ferritin and hemosiderin deposits as coarse blue granules, while dispersed ferritin is seen as diffuse faint blue cytoplasmic blush. Interestingly, ferritin deposition was found in the livers of our entire cohort of COVID-19 patients, both survived and deceased (Table 5). None of the patients showed absence of detectable iron residues; however, both the siderosis grading and the percentage of stained cells showed a trend of elevated values in livers of deceased patients (Figure 3C,D), compared to survived (Figure 3A,B) (Table 5).

Iron deposits were distributed into hepatocytes (Figure 3) (excluding the case of the obese patient, who also presented deposits in the Kupffer cells) and mostly found in cells in the area of liver acinus. Neither hepatocytes nor Kupffer cells showed iron deposits in control livers (Appendix A).

### 3.5. Ultrastructural Analysis

Transmission electron microscopy analysis confirmed the presence of iron accumulation and revealed fine details of iron deposits in livers of COVID-19 patients. All cases (both livers from survived and deceased patients) showed the presence of ferritin into hepatocytes, as demonstrated by the clear presence of electron dense particles (Figure 4). Abundant ferritin particles were found scattered in the cytoplasm (Figure 4A,B). Other particles were enclosed in siderosomes, membrane-delimited organelles derived from lysosomes (Figure 4A–D). These were frequently in the form of complex bodies, in which other material (such as lipids) was present (Figure 4B). In addition, siderosomes also presented hemosiderin granules, a form of deposits in which ferritin particles cannot be individually resolved (Figure 4C). Hemosiderin large aggregates were also not enclosed into siderosomes and were often found in strict contact with damaged mitochondria (Figure 4E,F), suggesting iron pathogenic role. To further evaluate liver injury, ultrastructural analysis was primarily performed on liver biopsies, to avoid misinterpretation due to the appearance of after-death changes on post-mortem specimens. Two striking features were found, fat accumulation and the presence of condensed mitochondria (Figure 5), which were clearly visible even by light microscope in resin-embedded semithin sections (Figure 5A,B). Transmission electron microscopy analysis showed the condensed state of mitochondria with compacted matrix and larger internal cristae (Figure 5C,D), a type of membrane remodeling observed with increased intracellular production of reactive oxygen species (ROS) and oxidative stress [15]. Of note, numerous mitochondria were in contact with lipid droplets (Figure 5C,D).

## 4. Discussion

Liver involvement in COVID-19 has been reported in severe cases of disease [1,16]. Elevation of hepatic enzymes as biochemical evidence of hepatitis has been reported, especially in patients with severe disease [17,18,19]. In critical COVID-19 patients, it has been proposed that liver dysfunction may occur as a consequence of hemodynamic alterations caused by cardiac failure and/or by mechanic ventilation, resulting in vascular pathology, endotheliitis and coagulopathy. However, histological examination of autoptic liver showed that not all cases presented hepatic vascular injuries [20]. Our results, obtained by CD34 immunostaining, indicative of vascular abnormalities, were associated with minimal congestion and dilatation of sinusoids and with mild fibrosis. Even a direct effect of SARS-CoV-2 on liver cells has not been proven with certainty. In our cases, PCR-test for SARS-CoV-2 performed on liver samples and immunohistochemical detection gave negative results; in addition, the electron microscopy analysis did not allow visualization of viral particles. SARS-CoV-2 uses the angiotensin-converting enzyme 2 (ACE2) and transmembrane protease serine 2 (TMPRSS2) for the entry into host cells [21]. ACE2 and TMPRSS2 are expressed in a variety of tissues including the liver; however, their expression extensively varies among the different cell types, being more highly expressed in cholangiocytes than in hepatocytes, and are expressed in the endothelial cells of blood vessels but not in the sinusoidal endothelium [22,23]. Hepatocyte and cholangiocyte organoids have been shown to be permissive to SARS-CoV-2 infection [24], but whether SARS-CoV-2 can infect hepatocytes and/or cholangiocytes in vivo is still debated [25], and recent data on in-depth proteomic assessment of autopsy tissue revealed little evidence of viral replication in the liver [26]. Therefore, the exact pathophysiological mechanisms to explain hepatic dysfunction in COVID-19 is presently not fully understood.

A significant increase in ferritin has been reported in moderate and severe COVID-19 patients [27]. The liver is a major site for body iron storage, deposited in form of complex ferritin protein molecules or hemosiderin [28]. Under physiological conditions, iron plays primary biological roles in oxygen transportation and in numerous enzymatic activities [29]. Nevertheless, excess iron causes oxidative tissue injury through free radical generation and oxidative modification of molecules [30]. Thus, iron overload may induce liver toxicity and cellular damage [31].

The aim of this study was to investigate the hypothesis that the liver of COVID-19 patients undergoes iron accumulation, providing a possible explanation for the occurrence of liver injury. Our results demonstrated that hepatocytes, in the liver of both survived and deceased patients, are sites of accumulation of both ferritin and hemosiderin, similarly to high-grade iron overload syndromes, when the capacity of ferritin to deposit iron decreases, and amounts of hemosiderin increases [32]. The results of semi-quantitative evaluation of hepatic iron underlined a trend to a more severe form of iron load in the liver of deceased patients, compared to survived, according to the emergent idea that imbalance of iron homeostasis may predict an adverse clinical course and increased mortality in COVID-19 [9]. Iron accumulation can drive hepatocytes to a form of cell death called ferroptosis, characterized by the presence of reactive oxygen species (ROS) and extensive lipid peroxidation [33]. ROS act as prime modulators of cellular dysfunction, contributing to many types of disease pathophysiology. In this regard, it has been recognized that oxidative stress is a major player in COVID-19 pathogenesis and severity [34]. Ferroptosis is morphologically characterized by the presence of ultrastructural changes in mitochondria, such as volume reduction and condensed mitochondrial membrane densities [35]. In our case, most mitochondria were in a condensed state, with compacted matrix and larger internal compartments, a configuration that is a manifestation of a functional state, with high oxidative capacity, coupled with marked phosphorylation [36]. Mitochondria play a central role in both processes of metabolism and oxidative stress; mitochondrial dysfunction and increased oxidative stress have been shown to play important roles in various human diseases [37]. We highlighted the association between iron deposits and mitochondrial damage. We also found an association between mitochondria and lipid droplets, suggesting that mitochondrial dysfunction may trigger LD biogenesis, by switching metabolic pathways to glycolysis and fatty acid biosynthesis [38]. In this regard, we also observed macrovesicular steatosis in all patients, despite the absence of underling predisposing conditions (except the patient with obesity). Interestingly, we previously described lipid droplets accumulation, mitochondria, and LD association in lungs from COVID-19 patients [39]. 

In conclusion, our results provide evidence that iron toxicity could be the cause of liver injury in COVID-19 patients. We hypothesize that iron loading of hepatocytes induces specific cellular damages, a pathogenic triangle of iron, mitochondria and lipids, which leads to liver damage. Despite the limitation represented by the small number of cases enrolled, this study enabled investigation of pathogenesis at the cellular level and described liver injury based on liver biopsies. Liver biopsies have rarely been performed on COVID-19 patients, but they are extremely important to understand the processes occurring in the liver at the time of the acute injury. A better understanding of the mechanisms contributing to liver injury will help to develop and implement early measures to prevent serious liver damage in patients suffering from COVID-19.

## Figures and Tables

**Figure 1 cells-10-01103-f001:**
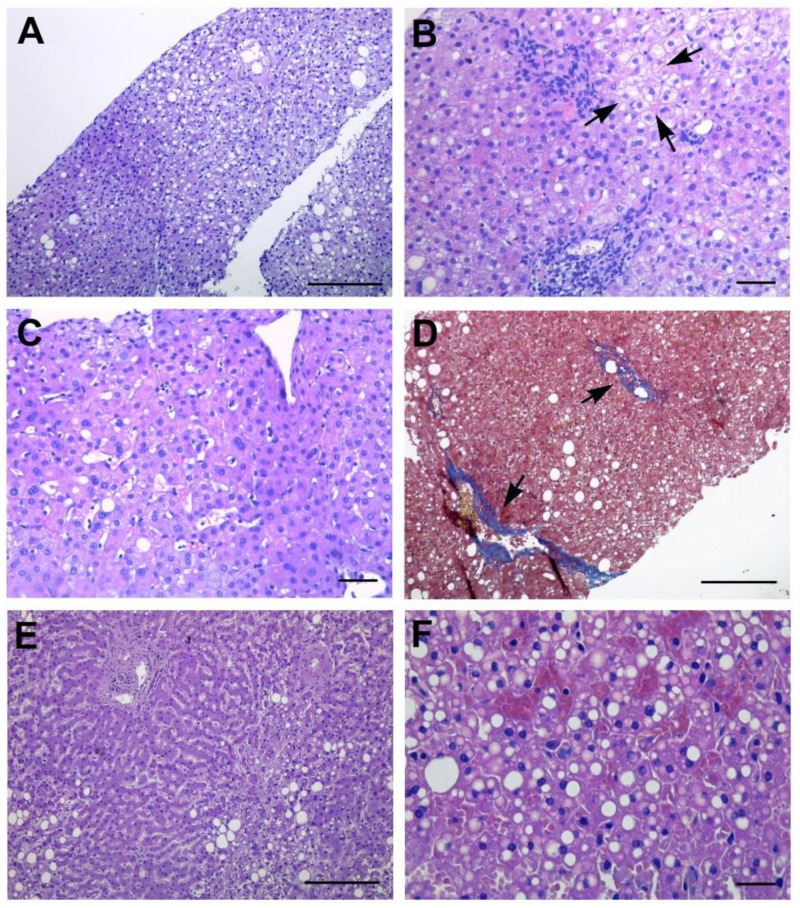
Histopathological features of liver biopsies and autoptic liver. (**A**–**D**) Liver biopsies show macrovesicular steatosis (**A**), and mild portal and lobular inflammation with spotty necrosis with presence of ballooning degeneration (arrows) (**B**). Sinusoidal dilatation in centrilobular areas is visible (**C**). Masson’s trichrome stain highlights mild fibrous expansion (arrows) (**D**). (**E**,**F**) Autoptic liver sections show macrovacuolar steatosis, and dilated but intact sinusoidal architecture; vascular fibrosis and spotty necrosis are visible (**E**). Hepatic sinusoidal dilatation and extravasation of red blood cells into the Disse’s space are visible; extensive steatosis is evident (**F**). Scale bars: **A**,**D**,**E** = 50 µm; **B**,**C** = 14 µm; **F** = 7 µm.

**Figure 2 cells-10-01103-f002:**
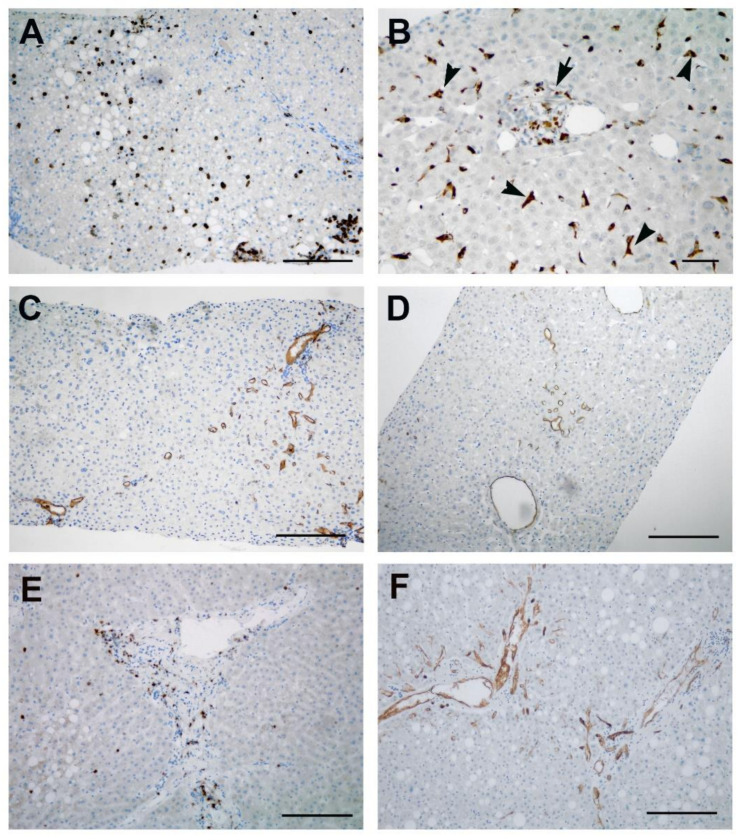
Immune-histochemical findings. (**A**–**D**) Immunostains of liver biopsies highlight the presence of numerous CD8+ T lymphocytes (**A**); CD68 labelling highlights the presence of macrophages in the portal areas (arrow), and numerous Kupffer cells into sinusoids (arrowheads) (**B**). CD34 immunolabeling shows a new angiogenesis process in periportal tracts (**C**) and centrilobular areas (**D**). (**E**,**F**) In autoptic liver sections, numerous CD8+ T lymphocytes are detected (**E**). CD34 positive staining is present in endothelial cells of the portal tract vasculature, and extending beyond the periportal regions with centrizonal microvessels and sinusoids positivity (**F**). Scale bars: **A,C,D**–**F** = 50 µm; **B** = 14 µm.

**Figure 3 cells-10-01103-f003:**
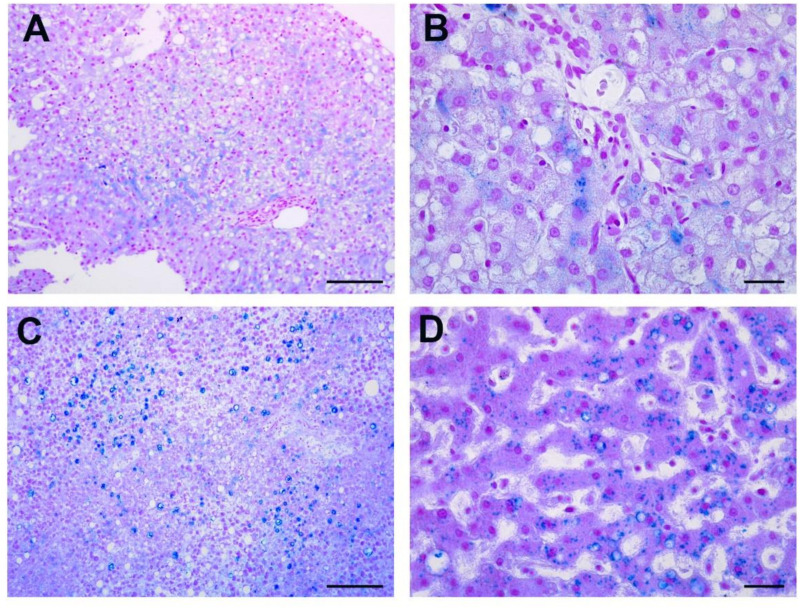
Iron deposits histological evaluation. (**A**,**B**) Perl’s staining of liver biopsies show coarse blue granules of ferritin and hemosiderin in hepatocytes. (**C**,**D**) Autoptic liver sections reveal heavy iron deposition in hepatocytes. Scale bars: **A**,**C** = 50 µm; **B**,**D** = 7 µm.

**Figure 4 cells-10-01103-f004:**
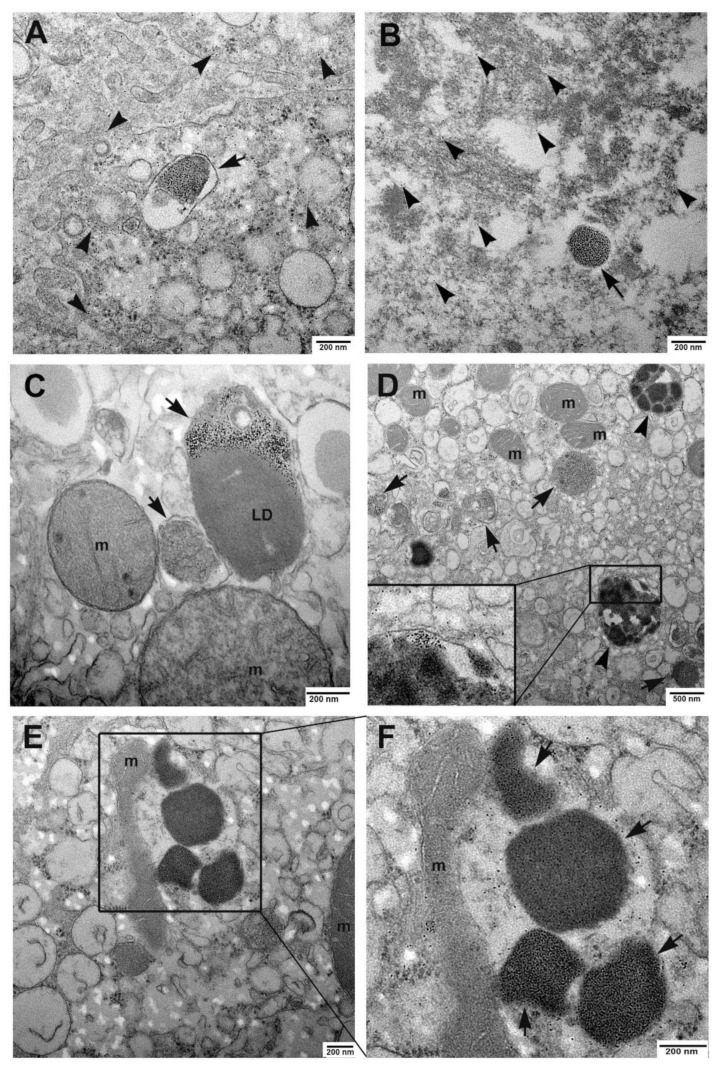
Ultrastructural analysis of iron accumulation in biopsies and autoptic livers. (**A**) Transmission electron microscopy of liver biopsies shows ferritin electron dense particles scattered in the cytoplasm of hepatocytes (arrowheads). Other particles are enclosed in siderosomes (arrow). (**B**) The analysis of autoptic liver sections reveals the presence of hemosiderin aggregates (arrow) and numerous ferritin particles diffuse in the cytoplasm. (**C**,**D**) Some siderosomes, beside ferritin particles, also contain lipid droplets or membranous structures (arrows); hemosiderin granules (arrowheads) are enclosed into organelles which also contain tiny ferritin particles (magnified boxed area in D). (**E**,**F**) Hemosiderin large aggregates are also found not segregated into siderosomes; higher magnification of the boxed area shows close contacts between hemosiderin aggregates (arrows) and a damaged mitochondrion; m, mitochondria; LD, lipid droplet.

**Figure 5 cells-10-01103-f005:**
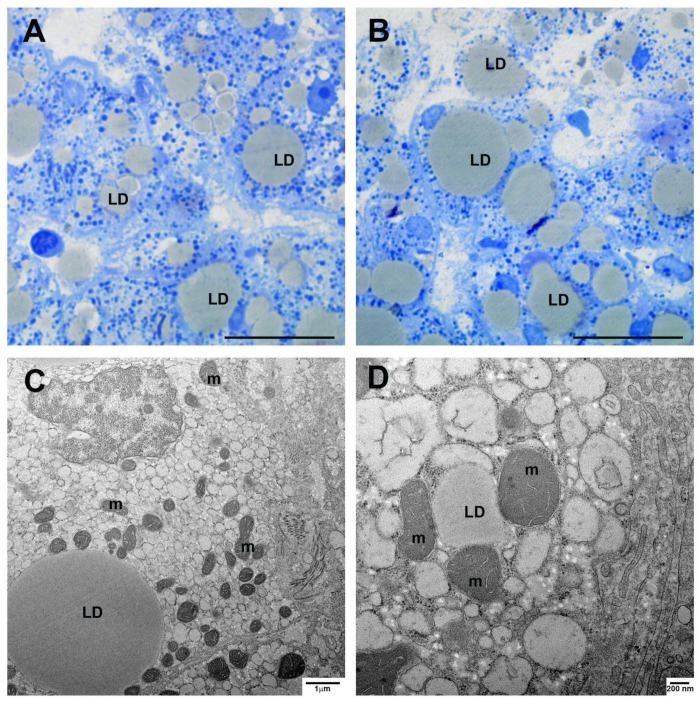
Lipid accumulation and mitochondria changes. (**A**,**B**) Light micrographs of liver biopsies semithin sections show the presence of abundant lipid droplets into hepatocytes and dark blue stained mitochondria. (**C**,**D**) The ultrastructural analysis highlights the condensed state of mitochondria, which display compacted matrix and enlarged cristae; mitochondria in strict contact with lipid droplets are visible (**D**). m, mitochondrion; LD, lipid droplet. Scale bars: **A**,**B** = 7 µm.

**Table 1 cells-10-01103-t001:** Demographics and clinical course of biopsied COVID-19 patients.

N°	Age/Sex	Comorbidities	Symptoms	Hospital Stay (Days)	Oxygenation/Intensive Care	Treatments
1	40/M	None	Fever; headache; myalgia and arthralgia; bilateral interstitial pneumonia	29	Oxygen therapy with VMK; cPAP	Lopinavir/ritonavir; Hydroxychloroquine; Enoxaparin
2	46/M	None	Fever; cough; myalgia and arthralgia; dyspnea; interstitial pneumonia	31	Oxygen therapy with VMK; NIV	Lopinavir/ritonavir; Hydroxychloroquine; Enoxaparin; Tocilizumab
3	42/M	Obesity	Fever; cough; dyspnea; bilateral interstitial pneumonia	41	Oxygen therapy with VMK 35%; NIV; cPAP orotracheal intubation	Lopinavir/ritonavir; Hydroxychloroquine; SarilumabCeftriaxone; doxycycline

NIV = not invasive ventilation; VMK = mechanical ventilation; cPAP = continuous positive airway pressure.

**Table 2 cells-10-01103-t002:** Demographics and clinical course of deceased COVID-19 patients.

N°	Age/Sex	Comorbidities	Symptoms	Hospital Stay (Days)	Oxygenation/Intensive Care	Treatments	Postmortem Causes of Death
1	76/M	Malignancy	Bilateral pneumonia	30	Oxygen therapy with cPAP	Tocilizumab	Cardiorespiratory failure.
2	69/F	Schizophrenia; staphylococcus	Fever; Dyspnea.	28	Oxygen therapy with VMK 35%	Not known	Cardiorespiratory failure.
3	63/M	Hypertension, cerebral ischemic vasculopathy	Fever; Dyspnea; Anosmia; Interstitial pneumonia.	5	Oxygen therapy with VMK 40%	Clarithromycin Rocephin, morphine; Rocuronium	DAD, cardiorespiratory failure.

NIV = not invasive ventilation; VMK = mechanical ventilation; cPAP = continuous positive airway pressure.

**Table 3 cells-10-01103-t003:** Laboratory characteristics of biopsied and autoptic COVID-19 patients.

Biopsied Patients N°	Laboratory Findings *									
	Temporal Trend	Fibrinogen	D-dimer	LDH	ALT	AST	Total Bilirubin	Ferritin	PCR	GGT
1	Admission	**915**	**815**	**345**	**120**	**82**	**1.48**	**1168**	**15.49**	**146**
	Peak during hospital	**525**	**513**	**479**	**747**	**330**	**0.75**	**1913**	**1.84**	**134**
	Discharge	**403**	**220**	**201**	**134**	**31**	**0.75**	**1056**	**0.07**	**66**
2	Admission	**883**	**603**	174	29	18	0.91	**488**	**11.84**	35
	Peak during hospital	**537**	**652**	215	481	102	0.66	**666**	**1.35**	185
	Discharge	**231**	**464**	269	28	21	0.68	**100**	<0.05	0.17
3	Admission	**915**	**803**	**626**	28	**60**	**1.3**	**1672**	**18.18**	**111**
	Peak during hospital	**1159**	**17,747**	**491**	137	**176**	**5.33**	**2028**	**36.77**	**760**
	Discharge	**382**	**1372**	**254**	100	**44**	**1.11**	**916**	**0.51**	**109**
**Autopsy Patients N°**										
1	Admission	**705**	**697**	**210**	20	31	0.41	**809**	**4.1**	18
	Peak during hospital	**139**	**17,180**	**>4500**	**>3300**	**>6000**	1.3	**1537**	**0.64**	125
	Death	**194**	**1069**	**3644**	**3068**	**2182**	0.86	**1407**	<0.05	55
2	Admission	253	**15,920**	**300**	19	**38**	0.39	**346**	**3.27**	12
	Peak during hospital	**17,575**	-	**337**	**83**	**64**	3.7	**1165**	**4.9**	147
	Death	**1242**	**44,554**	**386**	32	**19**	1.34	-	**0.37**	80
3	Admission	**648**	**1428**	-	**84**	**121**	0.57	-	**11.03**	39
	Peak during hospital	-	-	607	-	-	-	3292	-	-
	Death	**726**	-	-	**60**	**61**	0.4	-	-	53

* Normal Values: D-dimer ng/mL (0–500); fibrinogen mg/dL (150–450); LDH U/L (120–246 U/L); ALT U/L (0–55); AST U/L (5–34); total bilirubin mg/dL (0.2–1); ferritin μg/L (20–300); PCR mg/mL (<1 mg/mL); GGT U/L (woman > 38, men < 45).

**Table 4 cells-10-01103-t004:** Liver histopathological findings.

	Group BIOPSIES			Group AUTOPSIES		
	N° 1	N° 2	N° 3	N° 1	N° 2	N° 3
Steatosis	>5%	>33%	33–66%	33–66%	>5%	33–66%
Portal inflammation	Mild	Mild	Mild	Mild	Mild	Mild
Lobular inflammation	Focal *	Focal *	Focal *	−	Focal *	Focal *
Pattern of necrosis	−	Spotty	Spotty	−	Zonal	Zonal
Portal fibrosis	Mild	Mild	Moderate	Mild	Mild	Mild
Sinusoidal dilatation	+	−	+	+	+	+

* 2 to 4 foci per 10× objective. + present; − absent.

**Table 5 cells-10-01103-t005:** Iron presence in the liver.

	Group BIOPSY			Group AUTOPSY		
	n.1	n.2	n.3	n.1	n.2	n.3
Pattern of iron stain (Percentage of cells with iron deposits) *	Mild	Moderate	Moderate	Marked	Moderate	Marked
Siderosis (Hepatic iron graded according to Scheuer’s classification) #	1	2	3	4	3	4

* Percentage of stained cells was counted on Perl’s-stained liver sections. Minimal (<5%), Mild (5–33%), Moderate (34–67%), Marked (68–100%). # Siderosis was determined semi-quantitatively on Perl’s-stained liver sections according to Scheuer’s classification (Scheuer, PJ et al. J. Pathol. Bacteriol. 1962, 84, 53–64.): score 0, granules absent or barely discernible at a magnification of 400-fold (400×); 1, barely discernible at a magnification of 200× but easily confirmed at 400×; 2, discrete granules resolved at 100× magnification; 3, discrete granules resolved at a magnification of 25×; 4, massive granules visible even upon 10× magnification.

## Data Availability

All relevant data are within the manuscript.

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
