# Peer review of "Hepatic Failure in COVID-19: Is Iron Overload the Dangerous Trigger?"

_cells, 2021, doi:10.3390/cells10051103_

Round 1

Reviewer 1 Report

The authors have conducted a timely study to characterize the presence of ion deposition in human liver tissue section from survived and deceased patients with Covid19 infection.

Here are some minor but optional changes:

  1. For the purposes of those who are not completely familiar with details of normal liver pathology this reviewer suggests that figure 3 should include an image from normal patients using Perl’s staining. Since none of the patients showed absence of iron deposition this would be helpful.
  2. The authors mentioned that their liver samples did not have active infection. However, in the discussion section the authors should add a few lines about the expression of ACE2 and TMPRSS2 in different liver cells. For example, recent studies have show that liver organoids (including hepatocytes and cholangiocytes) express there proteins and that the virus can infect these cells – the author should add their discussion to give the reader a clearer idea of the cell types that the virus can infect.

Author Response

Referee #1

Here are some minor but optional changes:

For the purposes of those who are not completely familiar with details of normal liver pathology this reviewer suggests that figure 3 should include an image from normal patients using Perl’s staining. Since none of the patients showed absence of iron deposition this would be helpful.

ANSWER: As requested by the Referee we provided the results of Perl’s staining performed in control liver.  We add the new images in Supplementary Figure 1 of the revised version of the manuscript, in which also other immunohistochemical results performed in control livers were provided.

The authors mentioned that their liver samples did not have active infection. However, in the discussion section the authors should add a few lines about the expression of ACE2 and TMPRSS2 in different liver cells. For example, recent studies have show that liver organoids (including hepatocytes and cholangiocytes) express there proteins and that the virus can infect these cells – the author should add their discussion to give the reader a clearer idea of the cell types that the virus can infect.

ANSWER: We are grateful to the Referee’ suggestions that we took into account in the revised version of the manuscript. The liver expression of ACE2 and TMPRSS2 has been discussed in the text and new references have been added in the Reference list.

Reviewer 2 Report

The manuscript by Nonno et al., was a proposed research on the potential roles of the hepatic dysfunctions in COVID-19 disease. The authors focused on the post-mortem examination and tested the ferric related pathological mechanisms. The manuscript seemed to include unfinished work and therefore might be more suitable for more specific journals. There were some additional significant concerns:

  • Missing functional study.
  • Page 6 Lines 149-154 and related design, number per field should be counted in cell counting experiments.
  • Page 7 Lines 188-189, the “vascular remodeling”is unclear without analyzing the degree of the vessels; the alpha-SMA and vWF may be needed.
  • Missing essential controls, including the normal sample.
  • Page 9 Line 211, “predominantly in hepatocytes”if provide specific markers.
  • Lines 214-232, the missing evidence to show liver infection.
  • Need specific viral immuno-markers.

Author Response

Referee #2:

Reviewer's 2 criticisms are detailed below.

- Page 6 Lines 149-154 and related design, number per field should be counted in cell counting experiments.

ANSWER: We are grateful to the Referee’ suggestion. In the revised version of the manuscript the quantification of inflammatory cells has been performed as indicated in Materials and Methods, section 2.6. Results obtained have provided in the text of the revised version of the manuscript.

- Page 7 Lines 188-189, the “vascular remodeling”is unclear without analyzing the degree of the vessels; the alpha-SMA and vWF may be needed.

ANSWER: As requested by the Referee alpha-SMA and vWF immunostaining has performed. Results obtained have provided in the Supplementary Figure 2 of the revised version of the manuscript.

- Missing essential controls, including the normal sample.

ANSWER: We are grateful to the Referee indications. We improved the Materials and Methods information including the negative controls. In addition, we performed new immunohistochemical experiments on selected control liver samples, obtained from retrospective/archival samples. Results obtained were provided in Supplementary Figure 1 of the revised version of the manuscript.

- Page 9 Line 211, “predominantly in hepatocytes”if provide specific markers.

ANSWER: We thank the Reviewer to have highlighted this criticism. Histological identification of iron excess, is usually performed by Perl’s stain. Visual assessment of iron deposition can reveal three different patterns according to cellular distribution, that is: Parenchymal, characterized by iron deposition within hepatocytes; Mesenchymal, that corresponds to iron deposition within sinusoidal cells, mainly Kupffer cells; Mixed both in hepatocytes and Kupffer cells. The following images are provided as an example:

However, we understand that for those who are not completely familiar with details of liver pathology this point can be not so easily discernible. For this reason we provided two new images in Supplementary Figure 1, with Perl’s staining in control liver, to show absence of iron deposition in hepatocytes.

- Lines 214-232, the missing evidence to show liver infection.

 - Need specific viral immuno-markers.

 ANSWER: We apologize for the mistake, the sentence “livers of COVID-19 livers” has been amended as “livers of COVID-19 patients”. Indeed we had no prove of SARS-CoV-2 presence in the liver from our cases, in accordance with the negative results of SARS-CoV-2 PCR test performed on the liver tissues. For further investigation we had performed anti-coronavirus immunohistochemistry, utilizing two different antibodies, which resulted also negative, as shown in the images below.
